# The Bivalent Bromodomain Inhibitor MT-1 Inhibits Prostate Cancer Growth

**DOI:** 10.3390/cancers15153851

**Published:** 2023-07-28

**Authors:** Sanjeev Shukla, Carlos Riveros, Mohammed Al-Toubat, Jonathan Chardon-Robles, Teruko Osumi, Samuel Serrano, Adam M. Kase, Joachim L. Petit, Nathalie Meurice, Justyna Gleba, John A. Copland, Jay Chauhan, Steven Fletcher, K. C. Balaji

**Affiliations:** 1Department of Urology, University of Florida Health, Jacksonville, FL 32209, USA; 2Department of Neurological Surgery, University of Florida, Gainesville, FL 32611, USA; 3Department of Hematology Oncology, Mayo Clinic, Jacksonville, FL 32224, USA; 4Cancer Biology Department, Mayo Clinic, Jacksonville, FL 32224, USA; 5Department of Pharmaceutical Sciences, University of Maryland School of Pharmacy, Baltimore, MD 21201, USA

**Keywords:** bromodomain inhibitors, MT-1, prostate cancer, patient-derived xenografts, c-Myc

## Abstract

**Simple Summary:**

MT-1 decreased cell viability and causes cell cycle arrest in G0/G1 phase in castration sensitive and resistant PC cell lines in a dose-dependent fashion. The inhibition of c-Myc function by MT-1 was molecularly corroborated by de-repression of Protein Kinase D1 (PrKD) and increased phosphorylation of PrKD substrate proteins: threonine 120, serine 11 and serine 216 amino acid residues in β-Catenin, snail, and cell division cycle 25c (CDC25c) proteins respectively. This is first pre-clinical study demonstrating potential utility of MT-1 in the treatment of PC with c-Myc dysregulation.

**Abstract:**

Bromodomains (BD) are epigenetic readers of histone acetylation involved in chromatin remodeling and transcriptional regulation of several genes including protooncogene cellular myelocytomatosis (c-Myc). c-Myc is difficult to target directly by agents due to its disordered alpha helical protein structure and predominant nuclear localization. The epigenetic targeting of c-Myc by BD inhibitors is an attractive therapeutic strategy for prostate cancer (PC) associated with increased c-Myc upregulation with advancing disease. MT-1 is a bivalent BD inhibitor that is 100-fold more potent than the first-in-class BD inhibitor JQ1. MT-1 decreased cell viability and causes cell cycle arrest in G0/G1 phase in castration-sensitive and resistant PC cell lines in a dose-dependent fashion. The inhibition of c-Myc function by MT-1 was molecularly corroborated by the de-repression of Protein Kinase D1 (PrKD) and increased phosphorylation of PrKD substrate proteins: threonine 120, serine 11, and serine 216 amino acid residues in β-Catenin, snail, and cell division cycle 25c (CDC25c) proteins, respectively. The treatment of 3D cell cultures derived from three unique clinically annotated heavily pretreated patient-derived PC xenografts (PDX) mice models with increasing doses of MT-1 demonstrated the lowest IC_50_ in tumors with c-Myc amplification and clinically resistant to Docetaxel, Cabazitaxel, Abiraterone, and Enzalutamide. An intraperitoneal injection of either MT-1 or in combination with 3jc48-3, an inhibitor of obligate heterodimerization with MYC-associated protein X (MAX), in mice implanted with orthotopic PC PDX, decreased tumor growth. This is the first pre-clinical study demonstrating potential utility of MT-1 in the treatment of PC with c-Myc dysregulation.

## 1. Introduction

The c-Myc is overexpressed in almost three fourths of castration-resistant prostate cancer (CRPC), which is corroborated by the gain of chromosome 8q carrying the c-*Myc* gene [1]. Although associated with PC progression and a potential target for therapeutic intervention, c-*Myc* is not readily targetable due to a disordered tertiary helical structure, inaccessibility to the drug due to the predominant nuclear localization, and difficulty with targeted degradation [2,3,4]. Therefore, alternative approaches to c-Myc targeting in necessary and epigenetic targeting is an attractive strategy [5].

As changes at the epigenomic level are reversible, epigenetic therapies offer excellent opportunities, particularly in cancer, to reprogram aberrant cancer cells [6]. Several epigenetic drugs are in clinical practice to treat liquid cancers such as myelodysplastic syndromes and leukemia [7,8]. However, the efficacy of epigenetic drugs to date is limited in solid tumors and efforts are underway to improve epigenetic targeting [9]. One epigenetic target of particular interest is bromodomain (BD), which represents highly conserved genetic sequences that are involved in gene transcription through the recognition of histone acetyl-lysine post-translational modifications [10,11]. The c-Myc is known to play a role in chromatin remodeling, which is partly mediated through BET proteins [12]. There are four paralogs of the BET family of proteins, BRD2, BRD3, BRD4, and BRDT [13]. The aberrant c-Myc-driven oncogenic signaling induces hyperacetylation and BET family-mediated signal transduction and carcinogenesis [14]. The first generation BET BD inhibitor JQ1, has demonstrated preclinical efficacy in models of c-Myc-driven CRPC by targeting upstream of c-Myc, disrupting BRD4/androgen receptor (AR) interactions and the subsequent AR-directed transcription of c-Myc [1].

The low affinity for the BD binding pocket and short half-life of the first generation BD inhibitors, including JQ1, are unsuitable for clinical development [15], since several novel BD inhibitors have been developed including MT-1, which is 100-fold more potent in quantitative cellular assays than the parent compound JQ1 [16]. As opposed to JQ1, which represents a monovalent antagonist, MT-1 is an intramolecular bivalent BRD4 binder of both BD1 and BD2 [17]. MT-1 has shown promise in delaying the progression of leukemia in animal studies associated with an acceptable toxicity profile [16].

Our study demonstrates that BD inhibition by MT-1 is associated with decreased PC growth and viability in vitro and associated with an upregulation of PrKD expression. Animal studies in PDX mice models show that MT-1 is tolerated without dose limiting toxicity up to 4 weeks, which is promising for further testing for efficacy and clinical development. Moreover, our preliminary studies suggest MT-1 can be combined with drugs that inhibit c-Myc function through different mechanisms such as 3jc48-3, an inhibitor of obligate heterodimerization of c-Myc, with MYC-associated protein X (MAX) without increasing toxicity. The combinatorial strategies may be useful in future studies in enhancing the efficacy of MT-1 and improving the efficacy of c-Myc targeting. 

## 2. Methods

### 2.1. Reagents and Chemicals

MT-1 (BET–inhibitor) was procured from MedChemExpress (South Brunswick, NJ, USA), 3jc48-3 compound was synthesized in Dr. Steven Fletcher’s Laboratory at University of Maryland School of Pharmacy (Baltimore, MD, USA). MDV3100 was purchased from Millipore SIGMA (St. Louis, MO, USA).

### 2.2. Cell Viability

PC cell lines LNCaP (castration-sensitive) and PC-3 (castration-resistant), were procured from ATCC (American Type Culture Collection, Manassas, VA, USA). LNCaP cells were cultured in 10% fetal bovine serum (FBS), whereas, PC-3 cells were cultured in 5% with 1% penicillin and streptomycin (P&S) containing RPMI 1640 medium. The LNCaP and PC-3 cells were plated at a density of 1200 cells per well in 50 μL of RPMI 1640 media in 96-well plates for 24 h and treated with increasing dose of MT-1 drug (0.001, 0.01, 0.1, 1, and 10 μM) or 1% DMSO control (as vehicle) in 40 μL RPMI 1640 media for 72 h. In another experiment, PC cells were treated with 10 μM), 3jc48-3 (MYC/MAX dimerization inhibitor, MT-1 (0.1 μM), 3jc48-3 (10 μM), and MT-1 (0.1 μM) for 72 h. Following the incubation, we added 10 μL of Alamar Blue reagent (Thermo Fisher Scientific, Waltham, MA, USA) reagent to a final volume of 100 μL and further incubated for 4 h. Fluorescence was measured at 72 h after treatment, using Gen5 software version 3.14.03 package (BioTek, Singapore) on BioTek Synergy HT equipment. Protein absorbance was detected at 570 nm.

### 2.3. Cell Cycle Analysis

The LNCaP and PC-3 cells were seeded to 60–80% confluency in 24-well plates using the media as described above under the cell viability experiments. After 24 h, the cells were treated with MT-1 drug in triplicates at varying concentrations for 48 h. Based on the prior effective doses from the cell viability experiments, the LNCaP cells were treated at (0, 0.025, 0.050, and 0.1 µM) concentrations, whereas PC-3 cells were treated with (0, 0.1, 0.5, and 1 µM) concentrations of MT-1 drug. After 48 h of treatment, cells were trypsinized, collected in separate tubes, centrifuged, washed twice with PBS, fixed in 1 mL of 70% ethanol, and vortexed. Prior to cell cycle analysis, cells were re-washed twice with PBS and treated with DNase-free RNAse A (Sigma–Aldrich, St. Louis, MO, USA) to digest the RNA. Propidium iodide (PI) solution was used for staining the nuclei. Cells were incubated with 2 uL of PI (500 μg/mL) for 15 min at 37 °C, flow cytometric data were acquired using BD Accuri C6 Plus Cytometer (BD Biosciences, Franklin Lakes, NJ, USA) and analyzed using FlowJo version 10 software. All experiments were performed in triplicate. 

### 2.4. Protein Western Blot Analysis

Following varying concentrations and durations of MT-1 drug treatment of LNCaP and PC-3 cells like cell viability experiments, total cell lysates were prepared using RIPA Lysis and Extraction Buffer (Thermo Fisher Scientific), cytosolic and nuclear fractions of LNCaP and PC-3 cells separated. The cells were initially incubated in 250 µL ice-cold lysis buffer containing 10 mM HEPES (pH 7.9), 10 mM KCl, 0.1 mM EDTA, 0.1 mM EGTA, 1 mM DTT, 1 mM PMSF, and 0.5% NP-40 with freshly added protease inhibitors (leupeptin, aprotinin, and benzamidine) for 20 min. The cells were then scraped and the lysate was collected in a microfuge tube, mixed on a vortex and then centrifuged for 1 min (14,000× *g*) at 4 °C and stored in a −80 °C freezer. The supernatant was collected as cytosolic lysate and stored at −80 °C. The nuclear pellet was resuspended in 50 µL of ice-cold nuclear extraction buffer containing 20 mM HEPES (pH 7.9), 0.4 M NaCl, 1 mM EDTA, 1 mM EGTA, 1 mM DTT, 1 mM PMSF, and 0.5% NP-40 with freshly added protease inhibitors (leupeptin, aprotinin, and benzamidine) for 30 min with intermittent mixing. The tubes were centrifuged for 5 min (14,000× *g*) at 4 °C, and the supernatant (nuclear extract) was stored at −80 °C. After performing protein quantitation, 25 μg of protein was denatured using Laemmli buffer, heated for 5 min at 95 °C and loaded and resolved into 4–20% sodium dodecyl sulfate-polyacrylamide gels, transferred to a nitrocellulose membrane, blocked in 3% Bovine Serum Albumin (BSA) for 1 h at room temperature, and later probed using the specific primary antibody purchased from Cell Signaling Technology: PrKD1 (Cat #90039), c-Myc (Cat #9402), β-Catenin (#9562), Phospho-CDC25c (Cat #4901), Snail (#3895), and GAPDH (Cat #5174). The chemiluminescence reagent (ThermoFisher Scientific, Cat #34580, Waltham, MA, USA) was used to detect proteins. All experiments were performed in triplicate. Original blots can be found in Appendix A.

### 2.5. PDX 3D Spheroid Culture

We have successfully generated several patient-derived tumor xenograft (PDX) PC tumors in mice using PC tissue samples procured by biopsy of metastatic sites of patients heavily pre-treated with docetaxel, second generation anti-androgens including abiraterone, and/or enzalutamide. The PDX from these mice were harvested (GUR-017M generation 7, PRJ-88T generation 4, and TMA-027 generation 4), dissociated using MACS human tissue dissociation kit (Order no. 130-095-929) and gentleMACS^TM^ Octo Dissociated (Order no. 130-095-937) to produce single cells. Cells were plated in Corning Spheroid 96-well Microplate (Ref no. 4520) at a density of 3000–6000 cells per well to form spheroids, incubated at 37 °C and 5% CO_2_. Cell culture was maintained using proprietary DMEM-F12 50/50 (Corning ref. 15-090-CV)-based complete spheroid media. Cells were plated in triplicates. On day 5, drug treatment was performed using MT-1 at a concentration between 0.01 µM and 100 µM using 0.1% DMSO as a control. After 72 h of incubation, cell viability assay was performed using CellTiter-Glo^®^ 3D Cell Viability Assay (cat # G9681, Promega Corporation, Madison, WI, USA) and relative luminescence units (RLU) were obtained and coefficients of variation (CV) and Z-factor (Z’) were determined for quality metrics. A CV of <20% and a Z’ between 0.4 and 1.0 were determined acceptable for assay performance. The relative luminescence was normalized to internal plate controls and plotted against concentration (logarithmic scale) to show a dose response curve. Using GraphPad Prism 8.0, the 50% inhibitory concentration (IC_50_) with 95% confidence interval (CI) was calculated. 

### 2.6. Mice PDX and Xenograft In Vivo Studies to Test the Efficacy of MT-1, 3jc48-3 or MT-1 and 3jc48-3 Combination

The animal experiments were carried out following the Institutional Animal Care and Use Committee (IACUC) approval, study number 201910715. The aim of this experiment to assess the tolerability of MT-1 by mice and generate preliminary data on efficacy of MT-1 in reducing PC tumor growth. The c-Myc upregulated PC PDX engrafted mice were purchased from Jackson Laboratory, Ann Arbor, USA (# TM00298-upregulated functional c-Myc). The mice were administered the drugs intra-peritoneally (ip): vehicle control, or MT-1 (50 µg/kg) and/or 3jc48-3 (100 mg/kg) daily dose in vehicle (10% cremophor, 10% ethanol and 80% normal saline), 5 days a week for 4 consecutive weeks. To reduce the number of animals needed for the study, we refined the experimental design to include control animals from similar previous experiments with PDX tumor sizes of at least 0.5 mm^3^. Animal weights and tumor volume were measured on alternate days every week. We euthanized animals at the end of the experiment, collected and immediately stored tumor tissues in −80 °C freezer. Later, we homogenized the tumor tissues in RIPA lysis buffer and performed western blot analysis. 

## 3. Results

### 3.1. MT-1 Inhibits Growth of PC Cells and G0-G1 Cell Cycle Arrest In Vitro

To observe the prostate cell growth inhibitory effect, we treated castration-sensitive (LNCaP) and insensitive (PC-3) PC cells with (0.001 µM, 0.01, 0.1, 1 or 10 µM) of MT-1 for 72 h (Figure 1A,B). MT-1 significantly decreased LNCaP cells growth in a dose dependent fashion at 1 and 10 µM concentrations. PC-3 cells were resistant to low doses of MT-1 < 1 µM concentrations but the viability was significantly reduced at a higher dose of 10 µM at 72 h. The data suggest that MT-1 inhibits growth of LNCaP and PC-3 cells, albeit LNCaP cells being more sensitive to MT-1 inhibition at a lower concentration. MT-1 caused G0-G1 phase cell cycle arrest (Figure 1C,D) in LNCaP and PC3 cells within 48 h in a dose dependent fashion.

### 3.2. MT-1 Treatment of PC3 Cells Increased PrKD Protein Levels and Kinase Activity

Because we have previously shown that c-Myc is a transcriptional repressor of PrKD [18], we expected c-Myc inhibition by MT-1 to increase PrKD expression and kinase activity resulting in increased downstream substrate phosphorylation. Following treatment of PC-3 cells with MT-1 at concentrations of 0.1 and 0.5 µM for 24 h (Figure 1E), the total cell lysates demonstrated a 0.87-fold decrease and 1.2-fold increase in c-Myc and PrKD protein levels, respectively (Figure 1E). Consistent with an increase in PrKD expression, the known substrate phosphorylation of p-CDC25c (Ser-216), β-Catenin (T-120), and Snail (S-11) were increased from 1.11 to 2.5-fold, confirming an increase in PrKD kinase activity (Figure 1E). 

### 3.3. Combination with MT-1 and 3jc48-3 Results in Synergistic Inhibition of PC Cells’ Viability Associated with Increased PrKD Substrate Phosphorylation

We treated PC-3 and LNCaP cells with 3jc48-3 (10 µM), MT-1 (0.1 µM) or combination of 3jc48-3 (10 µM), and MT-1 (0.1 µM) for 72 h (Figure 2A,B). The LNCaP cells showed a significant decrease in cell viability after 3jc48-3 (9.42%) and MT-1 (7.26%) treatment. The treatment of LNCaP cells with 3jc48-3 and MT-1 decreased the viability by 18.06%, suggesting a synergistic effect. In PC-3 cells, while MT-1 was ineffective in reducing viability at 0.1 µM concentration, the addition of 10 µM 3jc48-3 significantly decreased the viability by 5.55%, similarly to 3jc48-3 alone, suggesting a lack of additive or synergistic effect in this castration-insensitive PC-3 cell line at the doses studied. The western blot analysis of PC-3 cells total protein lysates shows that the combination of 3jc48-3 and MT-1 is more effective in decreasing c-Myc and increasing PrKD proteins levels, and the phosphorylation of PrKD substrate proteins compared to either drug alone (Figure 2C). As PrKD1 expression has been shown to be repressed by androgens [19], we used MDV3100, an AR antagonist, as a positive control to upregulate PrKD1 expression.

### 3.4. MT-1 Decreased the Growth of 3D Tumor Spheroids Derived from PDX Tumors Generated from PC Metastatic Tissues Samples from Heavily Pre-Treated Patients

We cultured a total of three in vitro 3D spheroid cells using PC tumor tissue harvested from PDX mice models generated using clinically annotated tissue samples obtained by biopsy of metastatic sites of patients with advanced PC: GUR-017M, PRJ-88T, and TMA-027. The genetic mutation of each of the spheroids were characterized (Table 1). MT-1 inhibited the viability of all the three spheroids with the lowest IC_50_ of 0.04 µM for the GUR-017M (resistant to SGAA abiraterone and enzalutamide, and cabazitaxel) compared to 0.27 µM and 0.92 µM for PRJ-88T (resistant to enzalutamide and partially to docetaxel) and TMA-027 (resistant to enzalutamide) spheroids, respectively (Figure 3A–C). Interestingly, the most MT-1-sensitive spheroid GUR-017M also demonstrated c-Myc amplification, was most heavily pre-treated, and resistant to taxanes and SGAA. The results suggest the possible utility of MT-1 in heavily pre-treated patients with c-Myc amplification. 

### 3.5. MT-1 and 3jc48-3 Synergistically Reduce Tumor Growth Rate in PDX PC Mice Models

We purchased heterotopic c-Myc upregulated PDX PC models with tumors subcutaneously engrafted in the flank from Jackson Laboratories (Bar Harbor, ME, USA) and allowed the tumors to grow. The mice were randomly divided into four groups (N = three in control and four in each treatment group): Control, MT-1, 3jc48-3, or MT-1 + 3jc48-3. The mice were injected intraperitoneally with vehicle-only control or the drugs at a concentration of 3jc48-3 (100 mg/kg/day), MT-1 (50 μg/kg/day) or MT-1 (50 μg/kg/day) + 3jc48-3 (100 mg/kg/day), every day for 4 weeks. While the average animal weight during the study remained stable in MT-1 and MT-1+ 3jc48-3 groups, a mild decrease in animal weight was noted in response to 3jc48-3 treatment. There were no discernible toxicities in any of the animals. At the end of 4 weeks of treatment, we observed a significant decrease in tumor volume in all the three treatment groups compared to the control (Figure 4A). The tumor tissue samples harvested following 4 weeks of MT-1 treatment demonstrated increased PrKD1 protein levels associated with increased substrate phosphorylation of β-catenin (T-120) and Snail (S-11) compared to tumors from mice treated with vehicle-only control (Figure 4B).

## 4. Discussion

The c-Myc is a prolific proto-oncogene involved in several hallmarks of cancer [20], is dysregulated in three fourths of human cancers, and in about 70% of advanced PC but is yet to be successfully targeted [20]. The most successful strategy in the development of small molecular inhibitors in clinical medicine is targeting deep pockets of proteins with kinase activity that are often mutated in prostate cancers contributing to pro-oncogenic events [21]. The c-Myc protein, on the other hand, demonstrates none of the characteristics for successful small molecular targeting because of its disordered alpha helical monomeric structure, which makes it not conducive for designer drugs. Moreover, the c-Myc protein is often dysregulated but rarely mutated in PC and lacks enzymatic activity. The other class of drugs that have been successfully implemented in clinical practice are monoclonal antibodies that either target proteins on cell surface or their ligands [22]. As a transcription factor, c-Myc cannot be targeted by monoclonal antibodies either. Therefore, alternate options such as epigenetic BD inhibitors to effectively target c-Myc are extremely attractive, because c-Myc is regulated by BD-containing proteins and MT-1 as a BD inhibitor specifically targeting BD4 has been shown to down regulate c-Myc expression [14,16,23,24,25]. Our study provides pre-clinical evidence for the use of MT-1 in inhibiting PC growth.

MT-1 was generated following optimization of biochemical and biophysical properties of the parent first-in-class monovalent BD inhibitor JQ1. MT-1 leverages the tandem BD repeats in BET proteins and binds bivalently to BD domains. While the JQ1 and a bivalent ligand have similar dissociation constants for a single BD, bivalent compounds had profoundly greater effects on cell proliferation, supporting the conclusion that a functional avidity effect is at play in cellular contexts. A recent review article on the opportunities and challenges of BET inhibitors’ development highlighted the potential for combinatorial strategies. The most significant benefit of BET inhibition may be found when partnering with agents beyond those that affect AR, including immunotherapy, chemotherapy, radiotherapy, PI3K pathway inhibitors, poly (ADP-ribose) polymerase inhibitors, and combinations with other epigenetic drugs [26]. MT-1 showed a 400-fold improvement compared to JQ1 in activity in acute myeloid leukemia cells and highly prolonged exposure in vivo. MT-1 has a molecular weight of 1134 Da, which is above the conventional upper limit for small molecular inhibitors of 900 Da [16]. Further refinement in linker and structural chemistry may be needed prior to advancing MT-1 or its derivatives for clinical studies.

Several strategies are adopted in clinical practice to improve efficacy including combinations of drugs. For example, the addition of novel second-generation anti-androgens to standard androgen ablative treatments, such as medical or surgical castrations, has improved overall survival in men with castration-sensitive and resistant metastatic PC [27]. In this study, while MT-1 increased PrKD kinase activity presumptively due to de-repression by decreased c-Myc protein function, we were able demonstrate decrease in c-Myc function in the PDX tumor tissue, although we were unable to demonstrate lower protein levels of c-Myc despite MT-1 treatment in the PDX tumors. This finding may be a result of the limited quantitative sensitivity of western blot or insufficient efficacy of epigenetic targeting of c-Myc. We used an alternate strategy to improve the efficacy of c-Myc inhibition. The basic helical structure and zipper protein of c-Myc forms an obligate heterodimer with its partner MYC-associated factor X (MAX) to function as a transcription factor [18]. The addition of a second-generation small molecular MYC/MAX heterodimerization inhibitor 3JC48-3 to MT-1 enhanced the efficacy of PC growth inhibition in vitro and in vivo. The results are suggestive of potential therapeutic strategies that could be used to improve the efficacy of MT-1 and other c-Myc targeting agents. 

As c-Myc is a prolific transcription factor involved in a variety of normal cellular functions, a perceived risk of c-Myc targeting is systemic toxicity. We have previously demonstrated that 3JC48-3 as a novel c-Myc/MAX dimerization inhibitor, is tolerable in animal models with no change in either weight or signs predictive of adverse outcomes [18]. We postulated that this favorable side effect profile may be in part due to residual and necessary c-Myc transcriptional activity because of the partial inhibition of c-Myc/Max dimerization and binding to DNA activation sites, as the concentration required for complete inhibition may be too high [28]. Similarly, despite demonstrating reduction in c-Myc function, c-Myc proteins levels are not consistently reduced following MT-1 treatment, potentially contributing to critical c-Myc cellular functions and mitigating toxicity. The tolerability of MT-1 in animals with no apparent toxicity is consistent with previously published studies using c-Myc inhibitors and c-Myc mutant Omomyc [29]. The favorable toxicity profile of MT-1 in animal studies supports further development efforts for its clinical use. 

## 5. Conclusions

MT-1 is a bivalent BD inhibitor that inhibits PC growth in vitro and in vivo, and is associated with reduced c-Myc function including downstream PrKD kinase activity. The efficacy of MT-1 is enhanced by the addition of a MYC/MAX dimerization inhibitor. MT-1 is well-tolerated in mice at the doses studied with no apparent toxicity. The study supports the further pre-clinical development of MT-1 in PC.

## Figures and Tables

**Figure 1 cancers-15-03851-f001:**
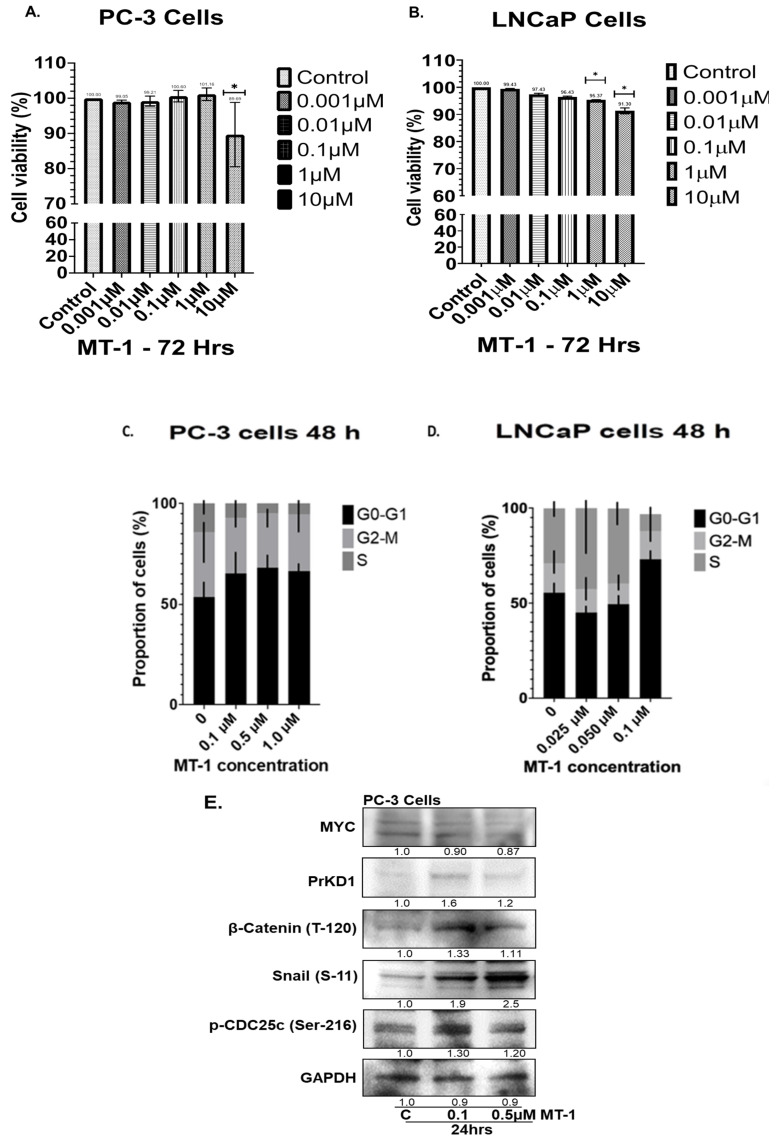
ATP cell viability assay: PC-3 (**A**) and LNCaP (**B**) cells were treated with MT-1 at varying concentrations (0.001–10 µM) for 72 h. MT-1 significantly decreased LNCaP cells growth in a dose dependent fashion at 1 and 10 µM concentrations of MT-1. PC-3 cells were resistant to low doses of MT-1 < 1 µM concentrations but the viability was significantly reduced at a higher dose of 10 µM MT-1 at 72 h (* indicates statistically significant compared to control, *p* < 0.05). Standard Error bars presented in the graph. The data suggest that MT-1 inhibits growth of LNCaP and PC-3 cells, albeit LNCaP cells being more sensitive to MT-1 inhibition at a lower concentration. Experiments were repeated in triplicate independently. MT-1 caused G0-G1 phase cell cycle arrest (**C**,**D**) in PC-3 and LNCaP cells, respectively, within 48 h in a dose dependent fashion. Western blot analysis of PC-3 cells with MT-1 at 0.1 and 0.5 µM concentrations for 24 h reduced and increased c-Myc and PrKD1 protein levels, respectively. Consistent with increased in PrKD expression, the known substrate phosphorylation of p-CDC25c (Ser-216), β-Catenin (T-120) and Snail (S-11) were increased from 1.11 to 2.5-fold, confirming an increase in PrKD kinase activity (**E**). The experiments were performed in triplicate independently. * *p*-value < 0.01.

**Figure 2 cancers-15-03851-f002:**
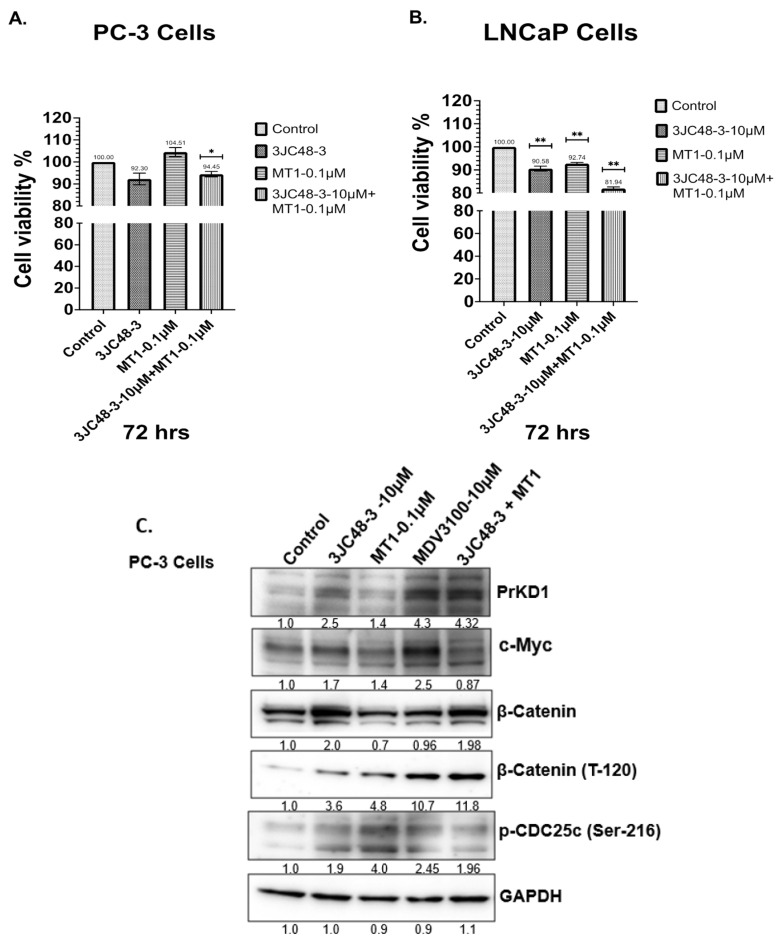
ATP cell viability assay: PC-3 (**A**) and LNCaP (**B**) cells were treated with either MT-1, 3JC48-3 or 3JC48-3 and MT-1 at varying concentrations for 72 h. The combination of 3JC48-3 and MT-1 was most effective in reducing cell viability compared to control or any of the single drug treatments (* indicates statistically significant compared to control, *p* < 0.05). Standard Error bars presented in the graph. Western blot (**C**) Compared to control, treatment of PC-3 cells with the drugs for 24 h reduced and increased c-Myc and PrKD1 protein levels, respectively. Consistent with increased in PrKD1 expression, the known substrate phosphorylation of p-CDC25c (Ser-216), β-Catenin (T-120), and Snail (S-11) were increased confirming an increase in PrKD kinase activity. Like viability assay, the combination of 3JC48-3 and MT-1 was most effective compared to single-drug treatments. As PrKD1 expression has been shown to be repressed by androgens, we used MDV3100, an AR antagonist, as positive control to upregulate PrKD1 expression. The experiments were performed in triplicate independently. * *p*-value < 0.01, ** *p*-value < 0.001.

**Figure 3 cancers-15-03851-f003:**
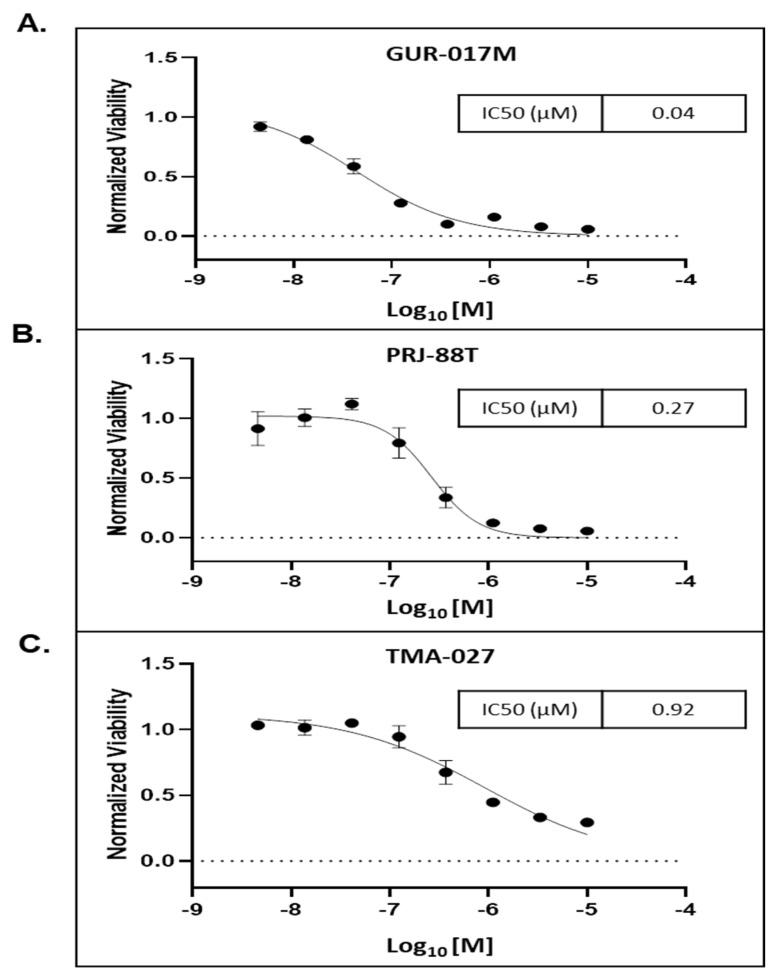
MT-1 compound effects on PC patient-derived 3D tumor spheroid culture: three-dimensional (3D) spheroid cell viability assay demonstrates MT-1 activity against three unique PC patient-derived tumor spheroids. (**A**) Metastatic castrate resistant PC GUR-017M patient tumor spheroid showing dose response growth inhibition at doses 0.5 µM to 10 µM, (**B**) metastatic castrate-resistant PC PRJ-88T patient tumor in spheroid showing dose response growth inhibition at doses 0.005 µM to 10 µM, and (**C**) metastatic castrate-resistant PC TMA-027 patient tumor in spheroid showing dose response growth inhibition at doses 0.01 µM to 10 µM of MT-1 compound. Standard Error bars presented in the graph.

**Figure 4 cancers-15-03851-f004:**
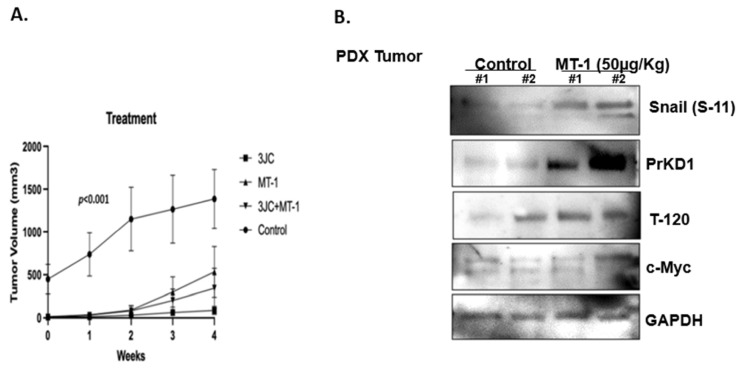
c-Myc inhibition by MT-1 and/or 3JC48-3 inhibits tumor growth in c-Myc upregulated patient-derived PC xenograft mice model by 4 weeks (**A**). Western blot analysis of tumor samples treated with MT-1 increased expressions of PrKD1 along with increased substrate phosphorylation of Snail (S-11) and β-Catenin (T-120). GAPDH is loading control for protein (**B**).

**Table 1 cancers-15-03851-t001:** Genetic mutation characteristics of spheroids: patient’s tumor-derived 3D spheroid characteristics are given in separate table representing prior and after drug biopsy and observed mutations in those spheroids. Experiments were performed in triplicates independently.

	Clinical Exposure Prior to Biopsy (Outcome)	Treatment after Biopsy (Outcome)	Mutations
**GUR-017M**	Abiraterone (resistant) Enzalutamide (resistant) Docetaxel (partial response) Cabazitaxel (resistant)	N/A	CCND1 amplification HRSA Q61R FGF 19, 4, 3 amplification LYN amplification MYC amplification TP53 splice site 672+2T>G
**TMA-027**	Enzalutamide (resistant) Docetaxel (partial response)	Enzalutamide (resistant) Cabazitaxel (partial response)	AR v7 Detected TP53 C.81/C>T NTRK1 c.2311 C>T
**PRJ-88T**	Enzalutamide (resistant)	Docetaxel + Pembrolizumab (partial response)	BRCA2 E1035 PTEN K263 PIK3R1 I405del SETD2 Q7

## Data Availability

Data generated or analyzed during the study are available from the corresponding author by request.

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
