# Peer review of "The Bivalent Bromodomain Inhibitor MT-1 Inhibits Prostate Cancer Growth"

_cancers, 2023, doi:10.3390/cancers15153851_

Round 1

Reviewer 1 Report

Dear Authors,

I read with interest your manuscript entitles: “The Bivalent Bromodomain Inhibitor MT-1 Inhibits Prostate 2 Cancer Growth”.

In a review of the role of BETs, the authors conclude that “the therapeutic potential of BET inhibitors for prostate cancer has been demonstrated in preclinical studies. However, further research is needed to identify biomarkers that can predict sensitivity to BET inhibitors and to develop novel, highly selective inhibitors to reduce toxicities. Finally, BET inhibitors are likely to hold the most clinical potential in combination with other agents.” It is, therefore, an article that I recommend reading and citing and using as food for thought for further discussion. Although it has been mentioned, this concept deserves further analysis. 

·      Mandl A, Markowski MC, Carducci MA, Antonarakis ES. Role of bromodomain and extraterminal (BET) proteins in prostate cancer. Expert Opin Investig Drugs. 2023 Mar;32(3):213-228. doi: 10.1080/13543784.2023.2186851. Epub 2023 Mar 9. PMID: 36857796.

The research was done properly, and the statistical analysis was done with rigor.

Author Response

18-July-2023

Prof. Dr. Samuel C. Mok,

Editor-in-Chief,

Cancers,

Subject: Submission of revised manuscript titled “The Bivalent Bromodomain Inhibitor MT-1 Inhibits Prostate Cancer Growth”.

Dear Reviewers,

We thank you for your time, efforts, and valuable insights. Thank you for giving us the opportunity to submit a revised draft of our manuscript titled: The Bivalent Bromodomain Inhibitor MT-1 Inhibits Prostate Cancer Growth . We have responded to all reviewers’ comments, incorporated  the changes in the revised manuscript and highlighted the changes within the manuscript. In addition, we have included point-by-point responses to the reviewers’ comments.  

Response to reviewer comments:

Reviewer 1:

In a review of the role of BETs, the authors conclude that “the therapeutic potential of BET inhibitors for prostate cancer has been demonstrated in preclinical studies. However, further research is needed to identify biomarkers that can predict sensitivity to BET inhibitors and to develop novel, highly selective inhibitors to reduce toxicities. Finally, BET inhibitors are likely to hold the most clinical potential in combination with other agents.” It is, therefore, an article that I recommend reading and citing and using as food for thought for further discussion. Although it has been mentioned, this concept deserves further analysis.

  • Mandl A, Markowski MC, Carducci MA, Antonarakis ES. Role of bromodomain and extraterminal (BET) proteins in prostate cancer. Expert Opin Investig Drugs. 2023 Mar;32(3):213-228. doi: 10.1080/13543784.2023.2186851. Epub 2023 Mar 9. PMID: 36857796.

Response: We agree with the reviewer’s comment. Based on his valuable suggestion we cited the aforementioned paper by Mandl et al. and further elaborated on this point in the final paragraph of the discussion section lines 7-13 in the revised manuscript.  

Reviewer 2:

The manuscript by Shukla et al. described the potentials of using BET inhibitor, MT-1, to reduce MYC function as a therapeutic strategy in PCa. MYC is a master transcription factor which is dysregulated in over 70% in cancers. However, due to the highly disordered protein structure, MYC is a challenging protein target. Using bromodomain inhibitors to target MYC is an alternative way to downregulate MYC protein level.  The following points need to be addressed:

  1. Line 46: “difficulty with targeted degradation because of prolonged half-life in physiological conditions.” MYC has a short half-life according to literatures (less than 30 min). References needed to support your claim.

Response:  We thank the reviewer for helping us to correct an error. Myc has a short half-life of 30 minutes much lower than > 4 hours for majority of proteins. We have corrected the error in the revised manuscript. We have deleted the statement “prolonged half-life in physiological conditions” in the introduction section, line 6 from the manuscript.

  1. Figure 1: The error bar of 10 µM MT-1 treated PC-3 cells was large. There were no error bars of DMSO controls in both lines. There were no error bars in cell cycle analysis. The significant figures were different in the western analysis.

Response: We regret the omission. We have made the suggested changes to Figure 1. 

  1. In the MT-1 treatment experiment, LNCaP cells were more sensitive. In the co-treatment with 3JC48-3, LNCaP cells showed synergistic effect while PC-3 showed no such effect. Why did you choose PC-3 cells over LNCaP cells for all the MYC, PrKD and substrate phosphorylation analysis?

Response: Based on our previous published work “PMCID: PMC2925514”. We have selected PC-3 cell line because PC-3 cells express high levels of endogenous c-Myc and low levels of PrKD1, whereas LNCaP cells have high levels of endogenous PrKD1. The PC-3 cells model allowed us to demonstrate the effect of c-Myc expression inhibition leading most effectively to increase PrKD1 expression and substrate kinase activity.

  1. Figure 2A and B: Keep the figure legends consistent. The significant figures were different in the western analysis.

Response: I thank the reviewer for this comment to improve the manuscript. Corrections made in figure 2A and 2B legends as suggested.  

  1. Figure 4: In 4A, the co-treatment was not as efficient as 3JC48-3 single treatment, any thoughts? In 4B, with MT-1 treatment, MYC protein was upregulated (1.9-fold compared to control), any explanation?

Response: Thank you for this insightful comment.  As shown in Fig 4A, MT-1 and 3JC48-3 significantly decrease tumor growth . However, 3JC48-3 is very effective in preventing tumor growth and hard to improve on the efficacy in this experimental model including combination with MT-1. It remains  unclear why the combination is worse with 3JC48-3 alone, but we speculate the drugs may interact with each other reducing their efficacy because of limited availability of the c-Myc target protein.  

We also agree with observation that c-MYC protein level is higher in the second tumor sample treated with MT-1. This may be due to tumor heterogeneity or experimental artifact. The GAPDH in the same column is also smudged compared to the first tumor sample. Nevertheless, we are encouraged that PrKD1 proteins levels are increased in both tumor samples engendering confidence that c-MYC is inhibited by MT-1.

  1. Line 318, Ref 26 only talked about CBP/EP300 and p53, which was irrelevant to the claim here.

Response: Thank you for pointing this out. We have removed the reference from the text.

In addition to the above comments, all spelling, grammatical and formatting errors have been corrected. We look forward to hearing from you in due time regarding our submission and to respond to any further questions and comments you may have.

Regards,

K.C. Balaji

Professor and Chair of Urology

Department of Urology

University of Florida, College of Medicine

Jacksonville, FL

Tel: 904-244-7340

Email: kc.balaji@jax.ufl.edu

Reviewer 2 Report

The manuscript by Shukla et al. described the potentials of using BET inhibitor, MT-1, to reduce MYC function as a therapeutic strategy in PCa. MYC is a master transcription factor which is dysregulated in over 70% in cancers. However, due to the highly disordered protein structure, MYC is a challenging protein target. Using bromodomain inhibitors to target MYC is an alternative way to downregulate MYC protein level.  The following points need to be addressed:

1.     Line 46: “difficulty with targeted degradation because of prolonged half-life in physiological conditions.” MYC has a short half-life according to literatures (less than 30 min). References needed to support your claim.

2.     Figure 1: The error bar of 10 µM MT-1 treated PC-3 cells was large. There were no error bars of DMSO controls in both lines. There were no error bars in cell cycle analysis. The significant figures were different in the western analysis.

3.     In the MT-1 treatment experiment, LNCaP cells were more sensitive. In the co-treatment with 3JC48-3, LNCaP cells showed synergistic effect while PC-3 showed no such effect. Why did you choose PC-3 cells over LNCaP cells for all the MYC, PrKD and substrate phosphorylation analysis?

4.     Figure 2A and B: Keep the figure legends consistent. The significant figures were different in the western analysis.

5.     Table 1 caption missed.

6.     Figure 4: In 4A, the co-treatment was not as efficient as 3JC48-3 single treatment, any thoughts? In 4B, with MT-1 treatment, MYC protein was upregulated (1.9-fold compared to control), any explanation?

7.     Line 318, Ref 26 only talked about CBP/EP300 and p53, which was irrelevant to the claim here.

8.     Incorrect figure captions formatting.

Minor points:

1.     Experimental section: Line 123: 80 ˚C, Line 168: 0.5 mm3

2.     IC50 instead of IC50

3.     MYC vs c-Myc

4.     Typo such as Line 283, Line 324…

Author Response

(The authors gave the same response as above.)

Round 2

Reviewer 2 Report

Thank you for addressing all the comments.